# Radiomics from Mesorectal Blood Vessels and Lymph Nodes: A Novel Prognostic Predictor for Rectal Cancer with Neoadjuvant Therapy

**DOI:** 10.3390/diagnostics13121987

**Published:** 2023-06-06

**Authors:** Siyuan Qin, Siyi Lu, Ke Liu, Yan Zhou, Qizheng Wang, Yongye Chen, Enlong Zhang, Hao Wang, Ning Lang

**Affiliations:** 1Department of Radiology, Peking University Third Hospital, 49 North Garden Road, Haidian District, Beijing 100191, China; 1510301315@pku.edu.cn (S.Q.); mnjb@live.cn (Y.Z.); wangqizheng96@163.com (Q.W.); chenyongye1995@163.com (Y.C.); 18811728521@163.com (E.Z.); 2Department of General Surgery, Peking University Third Hospital, 49 North Garden Road, Haidian District, Beijing 100191, China; 3Department of Radiology, Peking University International Hospital, Life Park Road No. 1 Life Science Park of Zhong Guancun, Chang Ping District, Beijing 102206, China; 4Department of Radiation Oncology, Cancer Center, Peking University Third Hospital, 49 North Garden Road, Haidian District, Beijing 100191, China

**Keywords:** rectal cancer, radiomics, neoadjuvant therapy, mesorectum, region of interest

## Abstract

The objective of our study is to investigate the predictive value of various combinations of radiomic features from intratumoral and different peritumoral regions of interest (ROIs) for achieving a good pathological response (pGR) following neoadjuvant chemoradiotherapy (nCRT) in patients with locally advanced rectal cancer (LARC). This retrospective study was conducted using data from LARC patients who underwent nCRT between 2013 and 2021. Patients were divided into training and validation cohorts at a ratio of 4:1. Intratumoral ROIs (ROI_ITU_) were segmented on T2–weighted imaging, while peritumoral ROIs were segmented using two methods: ROI_PTU_2mm_, ROI_PTU_4mm_, and ROI_PTU_6mm_, obtained by dilating the boundary of ROI_ITU_ by 2 mm, 4 mm, and 6 mm, respectively; and ROI_MR_F_ and ROI_MR_BVLN_, obtained by separating the fat and blood vessels + lymph nodes in the mesorectum. After feature extraction and selection, 12 logistic regression models were established using radiomics features derived from different ROIs or ROI combinations, and five–fold cross–validation was performed. The average area under the receiver operating characteristic curve (AUC) was used to evaluate the performance of the models. The study included 209 patients, consisting of 118 pGR and 91 non–pGR patients. The model that integrated ROI_ITU_ and ROI_MR_BVLN_ features demonstrated the highest predictive ability, with an AUC (95% confidence interval) of 0.936 (0.904–0.972) in the training cohort and 0.859 (0.745–0.974) in the validation cohort. This model outperformed models that utilized ROI_ITU_ alone (AUC = 0.779), ROI_MR_BVLN_ alone (AUC = 0.758), and other models. The radscore derived from the optimal model can predict the treatment response and prognosis after nCRT. Our findings validated that the integration of intratumoral and peritumoral radiomic features, especially those associated with mesorectal blood vessels and lymph nodes, serves as a potent predictor of pGR to nCRT in patients with LARC. Pending further corroboration in future research, these insights could provide novel imaging markers for refining therapeutic strategies.

## 1. Introduction

Colorectal cancer is the second most common cause of cancer–related deaths worldwide, with rectal cancer accounting for over one–third of all colorectal cancer cases [1]. Recently, preoperative neoadjuvant chemoradiotherapy (nCRT) has emerged as the standard treatment strategy for locally advanced rectal cancer (LARC), as it can increase the rate of local control and organ preservation [2,3]. However, the response to nCRT can vary widely, ranging from pathological complete response (pCR) with no remaining viable tumor cells to persistent disease (pathological no response, pNR) [4,5,6]. For those who show a poor response, extensive surgery may reduce the local recurrence rate of tumors and improve the prognosis [7,8]. A cohort study involving 1064 LARC patients revealed that patients with a poor response to nCRT had worse survival outcomes when the interval between the end of nCRT and surgery was longer [9]. Thus, predicting the treatment response before nCRT can lead to personalized treatment strategies, such as timely or extensive surgery, for patients who are less sensitive to nCRT.

However, a confirmation of nCRT response can only be obtained through the examination of resected specimens after surgery. Although previous studies have shown that several clinical factors and imaging assessments are associated with the response to nCRT in LARC patients [10,11,12,13,14,15], few clear and robust biomarkers have been identified to predict pathological responses [16]. Radiomics is a medical research method that uses computer technology to convert traditional radiological images into high–dimensional data for further analysis, thereby assisting in clinical decision–making [17]. Over recent years, radiomics has garnered substantial attention in the context of nCRT for LARC patients, with MRI–based radiomics anticipated to serve as a promising imaging biomarker capable of predicting treatment responses and prognostic outcomes [18,19,20,21]. However, it is worth noting that the majority of radiomics studies focus solely on extracting intratumoral radiomic features, neglecting the valuable information provided by the tumor microenvironment.

The extant literature suggests that peritumoral radiomic features in LARC are associated with various clinical outcomes, including response to neoadjuvant treatment, lymph node metastasis, neural invasion, disease–free survival (DFS), and overall survival (OS) [22,23,24,25]. This association could be attributable to the encapsulation of critical biological information, such as the patterns of tumor growth and metastasis, within peritumoral tissues [26]. However, most studies obtain the peritumoral region of interest (ROI) by dilating a specific distance outward from the tumoral ROI. This methodological approach presents two primary limitations: first, there is a lack of consensus regarding the optimal dilation distance; second, the inherent complexity of the peritumoral tissue components complicates the interpretation of radiomic features. Recent evidence suggests that the radiomic features of the mesorectal fat ROI correlate with the prognosis and treatment response of LARC [27]. This correlation may arise due to the integral role of mesorectal fat in facilitating the nutrient supply and catabolite drainage from the normal rectal wall and rectal tumors via vessels and lymphatics. Work by Kluza et al. [28] has shown that vascular parameters derived from dynamic contrast–enhanced (DCE) MRI are related to lymph node metastasis and response to nCRT. These findings collectively suggest that the radiomic features of mesorectal fat, vessels, and lymph nodes could serve as potential biomarkers for the treatment response in LARC. In light of these insights, this study proposed a novel method for ROI segmentation: the mesorectum is divided into ROI for mesorectal fat (ROI_MR_F_) and ROI for mesorectal vessels + lymph nodes (ROI_MR_BVLN_) based on signal differences in T2WI sequences. This approach, premised on the differentiation of tissue components, offers greater interpretability than traditional peritumoral ROI segmentation methods.

We hypothesized that both ROI_MR_F_ and ROI_MR_BVLN_ encompass crucial information pertinent to the prediction of the neoadjuvant treatment response. Thus, we aimed to investigate the value of radiomic features from intratumoral and different peritumoral ROIs for predicting the response to nCRT in LARC patients.

## 2. Materials and Methods

This retrospective study was approved by the institutional review board of Peking University Third Hospital, Beijing, PR China (IRB00006761–M2022474). The board waived the requirement for obtaining informed patient consent.

### 2.1. Study Patients

We included consecutive patients diagnosed with LARC who underwent nCRT between January 2013 and December 2021 at our hospital. The exclusion criteria were as follows: (1) distance from the anal verge greater than 10 cm; (2) not receiving standard nCRT or changing the nCRT regimen; (3) not undergoing TME surgery; (4) developing distant metastasis during the treatment period; (5) lacking pre–nCRT MRI data before treatment; (6) the absence of T2–weighted imaging (T2WI) sequences; (7) significant image artifacts that affect the determination of tumor boundaries; and (8) the absence of postoperative pathological results. The patient selection process is illustrated in Figure 1. A total of 209 patients were finally included in the study and were randomly assigned to a training cohort (167 patients) and a validation cohort (42 patients) at a ratio of 4:1.

### 2.2. Neoadjuvant Chemoradiotherapy

All patients received nCRT treatment according to a specific protocol. The radiation doses ranged between 45 and 50 Gy, administered over 25 fractions. The radiation clinical target volume encompassed the primary rectal cancer, perirectal and internal iliac nodes, mesorectum, pelvic sidewalls, and presacral space, with the upper edge at the sacral promontory. Concomitant oral capecitabine or XELOX regimen chemotherapy was administered during radiotherapy.

### 2.3. Reference Standard

The patients’ pathologic tumor regression grade (TRG) was assessed based on the American Joint Committee on Cancer (AJCC) eighth edition classification standard [4]. The TRG definitions used were as follows: TRG 0, indicating no tumor cells; TRG 1, indicating single tumor cells or small groups of tumor cells; TRG 2, indicating residual cancer with a desmoplastic response (mild regression); and TRG 3, indicating no tumor cells killed. In this study, we classified TRG 0–1 as a good pathological response (pGR) and TRG 2–3 as a poor pathological response (pPR).

### 2.4. MRI Protocol

All patients in our cohort underwent pretreatment rectal MRI consisting of standard high–resolution T2–weighted imaging (T2WI) on 3.0–T Discovery MR 750 (GE Medical Systems, LLC, 3200 N. Grandview Boulevard, Waukesha, WI, USA) or 3.0–T MAGNETOM Prisma (Siemens AG Healthcare, Erlangen, Germany). Detailed information regarding the parameters of the two scans is provided in Appendix A.

### 2.5. Image Segmentation

The oblique high–resolution T2WI sequence was used to perform the image segmentation for all patients. Digital Imaging and Communications in Medicine (DICOM) format images of each patient were uploaded to the uAI research portal (V1.1, United Imaging Intelligence, Co., Ltd., Shanghai, China) for analysis. Radiologist 1 (Q.S., with 3 years of experience in radiology) manually segmented all the regions of interest (ROIs), while radiologist 2 (L.K., with 3 years of experience in radiology) randomly selected 30% of all cases for re–segmentation to evaluate the inter–reader consistency. The two radiologists were kept unaware of the clinical and pathological information of the patients, as well as each other’s segmentation. Subsequently, an experienced radiologist (Z.Y., with 17 years of experience in abdominal radiology) reviewed and modified the ROIs segmented by radiologist 1. The modified ROIs were then utilized for the final analysis. After utilizing the uAI platform to adjust the image’s window width and level appropriately for a clear display of the lesion and mesorectum, we proceeded to segment the ROIs as outlined below:(1)Intratumoral region of interest (ROI_ITU_): First, the location and extent of the lesion was confirmed by combining the DWI and T2WI sequences. Subsequently, the tumor was manually segmented with meticulous attention to detail, ensuring its proper inclusion within the rectal contour and extension beyond the serosa, while simultaneously excluding any fibrous bands or spicules surrounding it.(2)The 2 mm peritumoral region of interest (ROI_PTU_2mm_) was generated by applying the “dilation” tool on the uAI platform to the initial ROI_ITU_, thereby expanding its boundaries by 2 mm and retaining the added portion. To ensure that the ROI solely consisted of the rectal wall and mesorectum around the tumor, the areas outside the mesorectal fascia, within the rectal lumen, and inside the tumor were manually excluded.(3)The 4 mm peritumoral region of interest (ROI_PTU_4mm_) was segmented using the same method as ROI_PTU_2mm_, except that the dilation distance was increased to 4 mm.(4)The 6 mm peritumoral region of interest (ROI_PTU_6mm_) was segmented using the same method as ROI_PTU_2mm_, except that the dilation distance was increased to 6 mm.(5)The mesorectal region of interest (ROI_MR_) refers to the area within the mesorectal fascia, outside the contours of the rectum and tumor, and below the peritoneal reflection.(6)The mesorectal fat region of interest (ROI_MR_F_) was created using the “threshold separation” tool on the uAI platform. The signal intensity threshold was adjusted to select only the fat signals (high signals) within the ROI_MR_, and a manual correction was carried out to remove the non–fat contents.(7)The mesorectal blood vessels + lymph nodes region of interest (ROI_MR_BVLN_) was created by adjusting the signal intensity threshold to select the middle to low signals within the ROI_MR_, which were mostly composed of the blood vessels and lymph nodes. A manual correction of the ROI was then performed to ensure that it only included blood vessels and lymph nodes.

The ROI segmentation process is shown in Figure 2.

### 2.6. Clinical and Follow–Up Information

Two weeks after segmentation, the clinical features comprising baseline information and MRI assessments were collected. The baseline information included age, gender, body mass index (BMI), the presence or absence of diabetes/hypertension, clinical T stage (cT), clinical N stage (cN), white blood cell count (WBC), hemoglobin level (HGB), platelet count (PLT), lymphocyte count, neutrophil count, eosinophil count, monocyte count, neutrophil–to–lymphocyte ratio, lymphocyte–to–monocyte ratio, platelet–to–lymphocyte ratio, carcinoembryonic antigen (CEA), cancer antigen 125 (CA125), and cancer antigen 199 (CA199). The MRI assessments consisted of the distance from the mass to the anal verge (DTAV), the tumor length, mesorectal fascia involvement (MRF), extramural vascular invasion (EMVI), and lateral pelvic lymph node metastasis (LPLN). MRF positivity was defined as the minimum distance between the tumor (mass, cancer nodule, metastatic lymph node, extramural vascular invasion, etc.) and the mesorectal fascia being ≤1 mm. EMVI positivity was defined as the presence of tumor signals within the blood vessels outside the rectal lumen where the tumor is located [29]. LPLN positivity was defined as the observation of pelvic enlarged lymph nodes (short axis >5 mm) outside the mesorectal fascia on MRI [30]. Appendix A shows a schematic diagram of the MRF, EMVI, and LPLN.

We followed up the patients by reviewing their inpatient and outpatient medical records and conducting phone interviews. Overall survival (OS) was defined as the duration from the surgery date to the latest follow–up or death caused by any reasons. Disease–free survival (DFS) was defined as the interval between the surgery date and the first incidence of local tumor recurrence or distant metastasi. If disease progression did not occur, DFS was determined as the period from the surgery date to the last follow–up.

### 2.7. Radiomics Feature Extraction and Selection

The uAI platform was utilized to perform image preprocessing and radiomic feature extraction. To reduce image heterogeneity, anisotropic pixels were resampled using B–spline interpolation to generate isotropic pixels of 1.0 × 1.0 × 1.0 (mm). The extracted features included first–order statistical features, shape features, texture features, and filter features, resulting in a total of 2264 features extracted from each ROI. The radiomic features generated are based on Pyradiomics [31], an open–source python package for the extraction of radiomic features from medical imaging. The definitions of all features can be found at https://pyradiomics.readthedocs.io/en/latest/features.html (accessed on 20 December 2022).

We evaluated the inter–rater reliability of the radiomic features extracted by two radiologists using the intra–class correlation coefficient (ICC). The ICCs ranges from 0 to 1, with a value between 0.80 and 1.0 indicating almost perfect agreement, 0.61 to 0.80 indicating substantial agreement, 0.41 to 0.60 indicating moderate agreement, 0.21 to 0.40 indicating fair agreement, and 0 to 0.20 indicating poor agreement. To ensure the extracted features were robust, only those with an ICC greater than 0.80 were included in subsequent analyses.

To make the features more comparable, we standardized the features of the training cohort using the z–score method: z = (X − X_mean_)/s. Here, X denotes the original feature value; X_mean_ and s represent the mean value and the standard deviation of the feature in the training cohort, respectively. Then we applied the same method to the validation cohort using the mean and standard deviation of the training cohort. Next, we removed features with a variance less than 1.0 using the variance threshold method. Then, we performed a statistical test on each feature and retained only those with a *p*–value less than 0.05. Finally, we employed the least absolute shrinkage and selection operator (LASSO) with five–fold cross–validation to select the features with the highest predictive power for pGR.

### 2.8. Model Construction

Radiomics models were developed using the logistic regression (LR) method, and their nomenclature followed the LR + subscript format based on the corresponding ROI. The models established using a single ROI comprised LR_PTU_2mm_, LR_PTU_4mm_, LR_PTU_6mm_, LR_MR_F_, and LR_MR_BVLN_. The models established using one intratumoral and one peritumoral ROI included LR_ITU+PTU_2mm_, LR_ITU+PTU_4mm_, LR_ITU+PTU_6mm_, LR_ITU+MR_F_, and LR_ITU+MR_BVLN_. Since there was no overlap between the mesorectal fat and mesorectal blood vessels and lymph node regions, a LR_ITU+MR_F+MR_BVLN_ model was also established. 

To train and validate the model and reduce the bias from data splitting, we used the five–fold cross–validation method. We randomly split 209 patients into five groups of similar sizes and proportions of pGR and pPR. We trained the model on four groups and validated it on the remaining one. We repeated this process five times so that every sample was in both the training and validation sets. We then evaluated the model’s performance using the average measures of the training and validation sets across all five groups.

The models’ performance was assessed by computing the average AUC of the validation cohort. The stability of the model was assessed by the coefficient of variation (CV) of the AUC in the validation cohorts of the five folds. Subsequently, for the model with the highest AUC, a radiomics score (radscore) was computed for each patient by utilizing the features and coefficients acquired from LASSO regression. The radscore was obtained using the following formula: radscore = β_0_ + β_1_X_1_ + β_2_X_2_ + β_3_X_3_ +···+ β_n_X_n_, where X_n_ refers to the nth selected feature and β_n_ denotes the coefficient associated with the nth feature. Furthermore, clinical models, radscore models, and clinical–radscore models (cli–radscore) were built and their performance was evaluated on the validation cohort.

### 2.9. Statistical Analysis

The categorical variables were analyzed using either the χ^2^ or Fisher’s exact test, while continuous variables were analyzed using either the independent–sample t–test (for normally distributed data) or the Wilcoxon rank–sum test (for non–normally distributed data). Variables with a *p*–value less than 0.1 in univariate analysis were subsequently included in a multiple stepwise logistic regression analysis, with the final selection of variables based on the Akaike’s Information Criterion (AIC) method. The AIC, which is a measure of the goodness–of–fit of a statistical model, was calculated as AIC = −2InL + 2k, where L represents the maximum likelihood of the model and k represents the number of adjustable parameters in the model. A smaller AIC value indicates a better fit of the model. Receiver operating characteristic (ROC) curves were plotted to quantify the differentiation performance of the established model, and the area under the curve (AUC) was calculated. DeLong’s test was utilized to compare any arbitrary two ROC curves. Moreover, decision curve analysis was employed to assess the clinical usefulness of each model. The “surv_cutpoint” function from the R package “survminer” was utilized to transform the continuous variables in survival analysis into categorical variables. Disease–free survival (DFS) durations were determined from the surgical date to either the occurrence of tumor recurrence/metastasis or the latest follow–up appointment. We designated the incidence of tumor recurrence/metastasis as an event, while patients who were lost to follow–up were considered censored. To visualize the DFS trends, we utilized Kaplan–Meier estimates to construct the DFS curves. Furthermore, the log–rank test was employed to scrutinize the differences across these curves. The Cox proportional hazards model was utilized for both the univariate and multivariate analyses to identify the risk factors associated with DFS. The predictive performance of the model for DFS was evaluated using the C–index. A two–tailed *p*–value < 0.05 was considered to indicate a statistically significant difference. All statistical analyses were performed with R software (version 4.2.0, http://www.Rproject.org, accessed on 20 December 2022) and SPSS (version 27.0, IBM, Armonk, NY, USA).

## 3. Results

### 3.1. Clinical Features

This study enrolled 209 consecutive patients with LARC, with 44 (21%), 74 (35%), 71 (34%), and 20 (10%) classified as TRG 0, 1, 2, and 3, respectively. Of these, 118 patients were classified as pGR, while 91 patients were classified as pPR. There were significant statistical differences (*p* < 0.05) observed in DTAV, tumor length, MRF, EMVI, platelet count, neutrophil count, NLR, and PLR between the two groups. No significant differences (*p* ≥ 0.05) were observed in the remaining clinical features (Table 1).

### 3.2. Feature Screening

A total of 2264 features were extracted from each ROI. Firstly, the inter–reader ICCs of radiomics features were calculated for each ROI. The mean ICCs were as follows: ROI_ITU_, 0.83; ROI_PTU_2mm_, 0.67; ROI_PTU_4mm_, 0.71; ROI_PTU_6mm_, 0.76; ROI_MR_F_, 0.76; ROI_MR_BVLN_, 0.76. Among the extracted features, the number of features with an ICC > 0.80 for ROI_ITU_, ROI_PTU_2mm_, ROI_PTU_4mm_, ROI_PTU_6mm_, ROI_MR_F_, and ROI_MR_BVLN_ were 1787, 930, 1186, 1553, 1417, and 1380, respectively. The distribution of ICCs for all 2264 features across each ROI is presented in Figure 3 and Appendix A.

Following the removal of features with an ICC ≤ 0.8, additional feature selection steps were carried out on the training cohort. Initially, the variance threshold method was utilized to eliminate the features with a variance < 1.0, which was followed by univariate feature selection to remove features with a *p*–value ≥ 0.05. Eventually, LASSO regression was implemented to select the features for modeling. Appendix A displays the count of the remaining features after each step of feature selection. Appendix A display the remaining features after each feature set selection.

### 3.3. Model Construction and Assessment

The models were developed using 10~28 features, and the logistic regression classifier was employed to establish the models. Five–fold cross–validation was performed. The results of the models, including the mean AUC, F1 score, sensitivity, specificity, and accuracy in both training and validation cohorts, are presented in Table 2. Several models were constructed using different combinations of intratumoral and peritumoral ROIs. Single ROI models included LR_ITU_, LR_PTU_2mm_, LR_P–TU_4mm_, LR_PTU_6mm_, LR_MR_F_, and LR_MR_BVLN_, with AUCs ranging from 0.689 to 0.79 in the validation cohort. Combined ROI models included LR_ITU+PTU_2mm_, LR_ITU+PTU_4mm_, LR_ITU+PTU_6mm_, LR_ITU+MR_F_, and LR_ITU+MR_BVLN_, and LR_ITU+MR_F+MR_BVLN_, with AUCs ranging from 0.79 to 0.859 in the validation cohort. The ROC curves for the different models are displayed in Figure 4. LR_ITU+MR_BVLN_ had the highest AUC among all the models. Among the 28 features used to establish the LR_ITU+ MR_BVLN_ model, 15 were derived from ROI_ITU_ and 13 were from ROI_MR_BVLN_. The LASSO regression result and feature coefficients are shown in Appendix A. The radscore was calculated by adding up the product of each feature value and its corresponding coefficient.

### 3.4. Clinical, Radscore, and Cli–Radscore Models

In both the training and validation cohorts, the radscore of the pGR group was higher than that of the pPR group (Figure 5a,b, *p* < 0.001). A multivariate stepwise logistic regression analysis was conducted on clinical factors with a *p*–value less than 0.1, revealing that two factors, namely DTAV > 4 cm (OR = 4.41, 95%CI 2.04–9.52, *p* < 0.001) and PLR (OR = 0.99, 95%CI 0.99–1.00, *p* = 0.008), were independent predictors of pGR. The results of the univariate and multivariate analyses are illustrated in Appendix A. The clinical model had an AUC of 0.702 and 0.618 in the training and validation cohorts, respectively. In comparison, the radscore model had an AUC of 0.913 and 0.884, and the cli–radscore model had an AUC of 0.924 and 0.873 in the training and validation cohorts, respectively. The ROC curves for clinical, radscore, and cli–radscore models are presented in Figure 5c,d. The decision curves for the clinical, radscore, and cli–radscore models in both the training and validation cohorts are shown in Figure 5e,f. The overall findings suggest that incorporating the clinical factors does not improve the predictive ability and clinical applicability of the radscore for pGR.

### 3.5. The Association between Radscore and Disease–Free Survival

The follow–up period for the study participants ranged from 2 to 123 months, with a median duration of 42 months and a mean duration of 45.6 months. Within this period, 42 patients (18.3%) experienced disease progression, with 10 cases (4.4%) showing local recurrence and 38 cases (16.6%) presenting distant metastasis. Specifically, lung metastasis was observed in 17 cases, liver metastasis in 16 cases, bone metastasis in 4 cases, lymph node metastasis in 3 cases, and metastasis of unknown location in 2 cases. Moreover, seven patients died during the follow–up period.

Appendix A displays the optimal cut–off values for each variable, alongside their corresponding log–rank statistics. These values were determined by utilizing the “surv_cutpoint” function within the R package “survminer” to convert continuous variables into categorical ones. To analyze the 5–year DFS of patients, any follow–up durations exceeding 60 months were uniformly recorded as “60 months.” In the high radscore group, the median follow–up duration was 38.5 months, with recurrence/metastasis observed in eight patients. Meanwhile, in the low radscore group, the median follow–up duration stood at 25 months, and recurrence/metastasis was witnessed in 34 patients. A log–rank test demonstrated that patients with a high radscore tend to achieve a more extended DFS (*p* = 0.029) (Figure 6a). We conducted univariate and multivariate Cox analyses on variables with *p*–values less than 0.05 from all log–rank tests. Table 3 presents the results of these analyses, as well as the corresponding C–indices for predicting the 5–year DFS. Figure 6b displays the forest plot of the multivariable Cox regression results.

## 4. Discussion

In this study, we established several models based on radiomic features of the intratumoral ROI and different peritumoral ROIs from pre–nCRT MRI. The model that combined features of ROI_ITU_ and ROI_MR_BVLN_ had the highest AUC of 0.859, with a sensitivity of 78.9%, specificity of 80.3%, and accuracy of 79.4%. This model could accurately distinguish between pGR and pPR patients and outperformed other combinations of tumoral and peritumoral ROIs. Notwithstanding the substantial differences in numerous clinical variables—such as DTAV, tumor length, MRF, EMVI, platelet count, neutrophil count, NLR, and PLR—observed between the pGR and pPR groups in our patient data comparisons, only DTAV and PLR retained their statistical significance following univariate and multivariate analyses on the training cohort. This suggests potential instability in the clinical factors under consideration. Additionally, our findings revealed that incorporating these clinical factors did not bolster the predictive accuracy of the radscore model. A plausible interpretation might be attributed to the inherent volatility of the clinical factors. These factors within an individual patient may demonstrate variability over time, be subjected to alterations across diverse physiological conditions, or fall prey to inaccuracies in measurement, culminating in unpredictable predictive outcomes. For example, the appraisal of radiological characteristics such as DTAV, tumor length, MRF, and EMVI can be subject to the physician’s expertise or inaccuracies in measurement. Moreover, hematological parameters including platelet count, neutrophil count, NLR, and PLR may undergo fluctuation, depending on the overall immune–inflammatory status of the patient.

In the realm of MRI for rectal cancer, high–resolution T2WI sequences are predominantly utilized, offering a vivid delineation of both the rectal neoplasm and its adjacent structures [32,33]. T2WI is the most commonly employed sequence for rectal cancer radiomics, followed by the diffusion–weighted imaging (DWI) or apparent diffusion coefficient (ADC) sequence [34]. Within the context of this study, the DWI sequence was consciously omitted owing to its inferior resolution, a characteristic found to impede radiologists in executing effective ROI segmentation. Shin et al. demonstrated that combining T2WI and DWI did not enhance the predictive performance of radiomic models for pCR compared to using T2WI alone [19]. Moreover, we opted to solely use T2WI in this study to reduce the time and effort required for segmentation by radiologists, thereby facilitating the translation of radiomic models from theory to clinical applications. Nevertheless, in our study, only manual segmentation was undertaken. It was found in Defeudis et al.’s research that models perform more effectively in external validation sets when automatic segmentation is employed [35]. Given the similarity of signals in rectal mesenteric fat, blood vessels, and lymph nodes on T2WI, they are more readily identifiable by artificial intelligence. Hence, in comparison to tumors possessing complex signals, the prospect for successful automatic segmentation is significantly enhanced.

Radiomics research has recently focused on the peritumoral ROI due to its potential to provide valuable information about the tumor microenvironment, which can aid in assessing treatment efficacy and predicting tumor prognosis. However, a standardized definition of the peritumoral ROI is currently lacking. Most studies define it as the area surrounding the tumor within a certain distance, but determining the optimal distance remains a challenge. For example, in a study predicting the prognosis of non–small cell lung cancer, the peritumoral ROI was defined as the area 15 mm outside the lesion [36]. In another study on the postoperative recurrence of liver cancer, two peritumoral ROIs were defined: the micrometastasis area (0–1 cm) and the potential cirrhosis background (1–2 cm) around the tumor [37]. A study predicting the grading of renal clear cell carcinoma defined the peritumoral region as the area 2 mm, 5 mm, and 10 mm around the tumor, further dividing these regions into the peritumoral parenchyma and peritumoral fat. The results showed that the radiomic features of the peritumoral fat contained important predictive information [38]. However, our study found that the inclusion of peritumoral ROIs at 2 mm, 4 mm, or 6 mm did not significantly enhance the performance of the radiomics model beyond the use of only the intratumoral ROI. In a separate study by Pizzi et al. [39], the radiomic features from pretreatment MRI were utilized to predict the pCR after nCRT in 72 patients with LARC. The radiomic features were extracted from ROIs corresponding to the tumor core (TC) and tumor boundary (TB). The results showed that the model incorporating TC, TB, and clinical features achieved an AUC of 0.793, which was significantly higher than the AUCs of the models using TC + clinical features or TB + clinical features alone (0.689 and 0.541, respectively). This finding contrasts with our own results, and one plausible explanation is that rectal tumors display heightened activity levels and are surrounded by a complex composition of tissues. Hence, it is imperative to identify additional peritumoral radiomics biomarkers. We segregated the mesorectum into two ROIs based on distinct tissue components: mesorectal fat (ROI_MR_F_) and mesorectal blood vessels and lymph nodes (ROI_MR_BVLN_). Our findings revealed that the model (AUC = 0.859) that integrated features from ROI_ITU_ and ROI_MR_BVLN_ outperformed the models employing ROI_ITU_ and ROI_MR_BVLN_ separately. In this model, we utilized 15 texture features extracted from ROI_ITU_, alongside 4 first–order features from ROI_MR_BVLN_ and additional 9 texture features from ROI_MR_BVLN_. As an example, the “Original_GLSZM_HighGrayLevelZoneEmphasis” texture feature extracted from ROI_MR_BVLN_ can provide insights into the distribution of high–intensity pixels in an image. This feature is indicative of the co–occurrence strength of the high gray–level values within the image, allowing for the identification of bright or light areas present in the image. A higher value of this feature corresponds to a greater prevalence of bright areas in the image. By comparing the values of this feature, we can discern differences in the signal intensity distribution between pGR and pPR patients. Our results indicated that the tumor, blood vessels, and lymph nodes in the rectal vicinity all harbored complementary information linked with response to nCRT. The radscore, predicated on the fusion of ROI_ITU_ and ROI_MR_BVLN_ features, also correlated with the treatment response and prognosis of LARC patients post–nCRT. The radiomic features of the blood vessels and lymph nodes around the tumor could be potential imaging biomarkers for predicting the response to nCRT, potentially due to the following reasons: firstly, the growth and nutrient uptake of tumors are closely linked with microvascular density around the tumor; secondly, hematogenous and lymphatic metastasis are the principal pathways of rectal cancer metastasis; thirdly, radiotherapy per se affects the development of the microvessels around the tumor. These reasons could lead to morphological and signal alterations of blood vessels and lymph nodes, which are intimately linked with tumor growth and metastasis, and could be reflected in the radiomic features [40,41].

The present study has several limitations that should be acknowledged. Firstly, it is a single–center retrospective study, which may lead to a potential selection bias, as we did not consider patients who underwent conservative treatment. Secondly, external validation was not performed, and therefore the generalization performance of the model requires further verification. Thirdly, although we segmented the region of interest based on different tissue components, the relationship between radiomic features and histopathological physiology remains unclear, and additional studies are needed to establish a connection between these features and biological behavior. Fourthly, the manual segmentation of ROI may lead to inter–reader variability, but we attempted to ensure the robustness of the features using ICC analysis, and we plan to develop automatic segmentation models as a future research direction.

## 5. Conclusions

In conclusion, our findings validated that the integration of intratumoral and peritumoral radiomic features, especially those associated with mesorectal blood vessels and lymph nodes, serves as a potent predictor of pGR to nCRT in patients with LARC. Additionally, the radscore, extrapolated from this model, demonstrated a notable correlation with the duration of DFS following surgery in the patient cohort. Pending further corroboration in future research, these insights could provide novel imaging markers for refining therapeutic strategies.

## Figures and Tables

**Figure 1 diagnostics-13-01987-f001:**
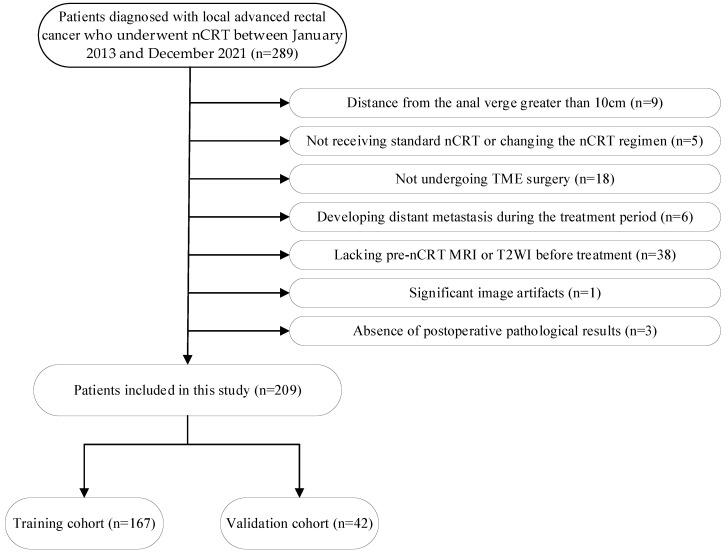
The patient selection process.

**Figure 2 diagnostics-13-01987-f002:**
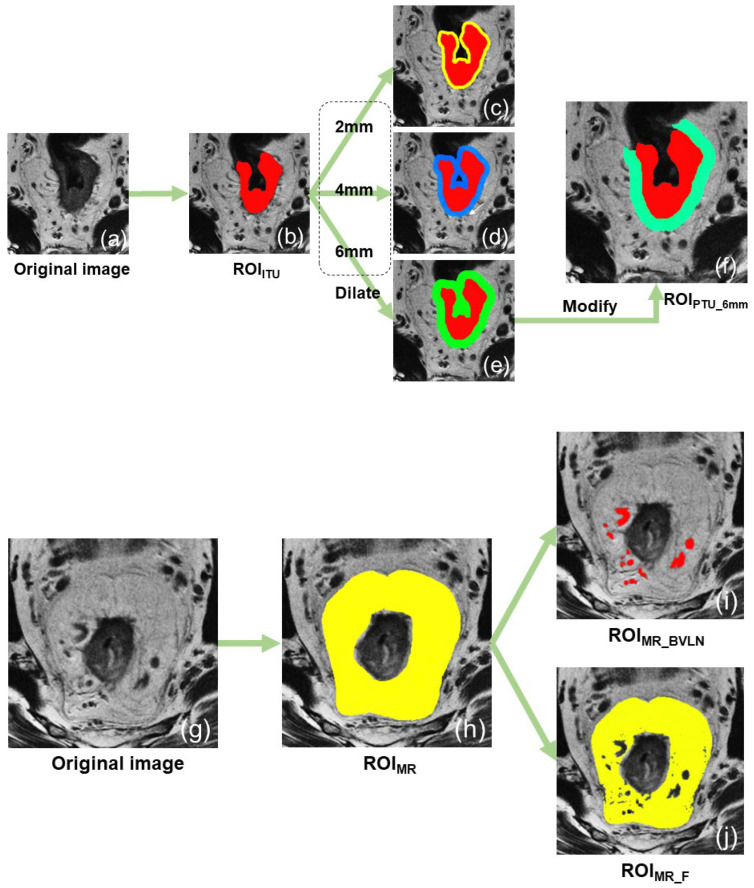
The process of segmenting the ROIs. (**a**,**g**) Oblique axial high–resolution T2–weighted images. (**b**) ROI_ITU_. (**c**–**e**) The ROIs obtained by dilating ROI_ITU_ by 2 mm, 4 mm, and 6 mm outward, respectively, and keeping only the different parts. (**f**) ROI_PTU_6mm_, which was manually adjusted based on (**e**) to include only the rectal wall and mesorectal fat around the tumor. (**h**) ROI_MR_. (**i**,**j**) Two ROIs derived from (**h**) by separating it based on the signal intensity threshold, and manually adjusted to find the ROI_MR_BVLN_ and ROI_MR_F_. ROI, region of interest.

**Figure 3 diagnostics-13-01987-f003:**
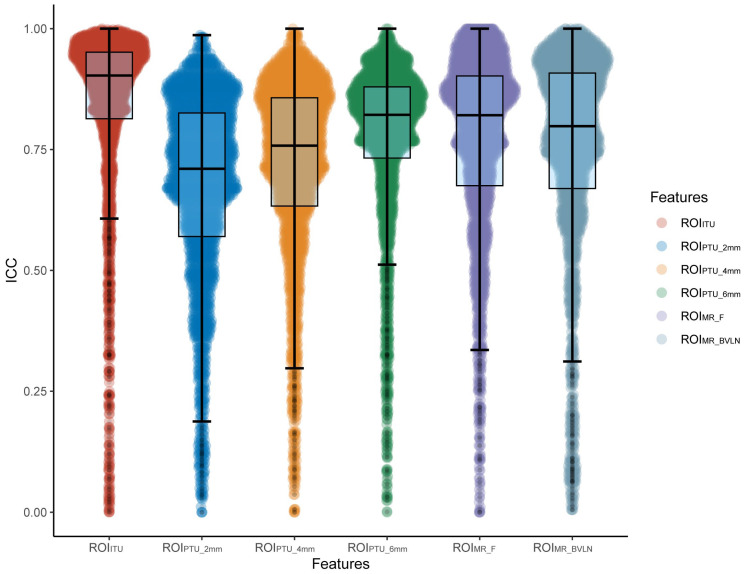
ICC boxplots and violin plots for each ROI. The boxplots displayed the median, first quartile, third quartile, and range of outlier values for each group of ICCs, while the violin plots showed kernel density estimation curves for the magnitude of ICCs for each group of features. ROI, region of interest; ICC, intraclass correlation coefficient.

**Figure 4 diagnostics-13-01987-f004:**
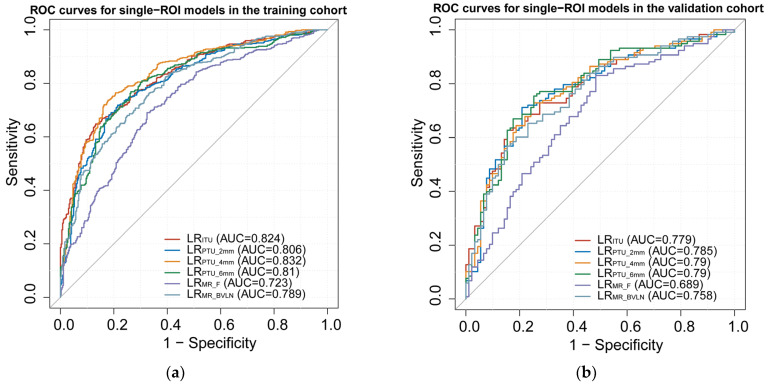
ROC curves for the single–ROI models in the training (**a**) and validation (**b**) cohorts. ROC curves for the combined–ROI models in the training (**c**) and validation (**d**) cohorts. ROC, receiver operating characteristic.

**Figure 5 diagnostics-13-01987-f005:**
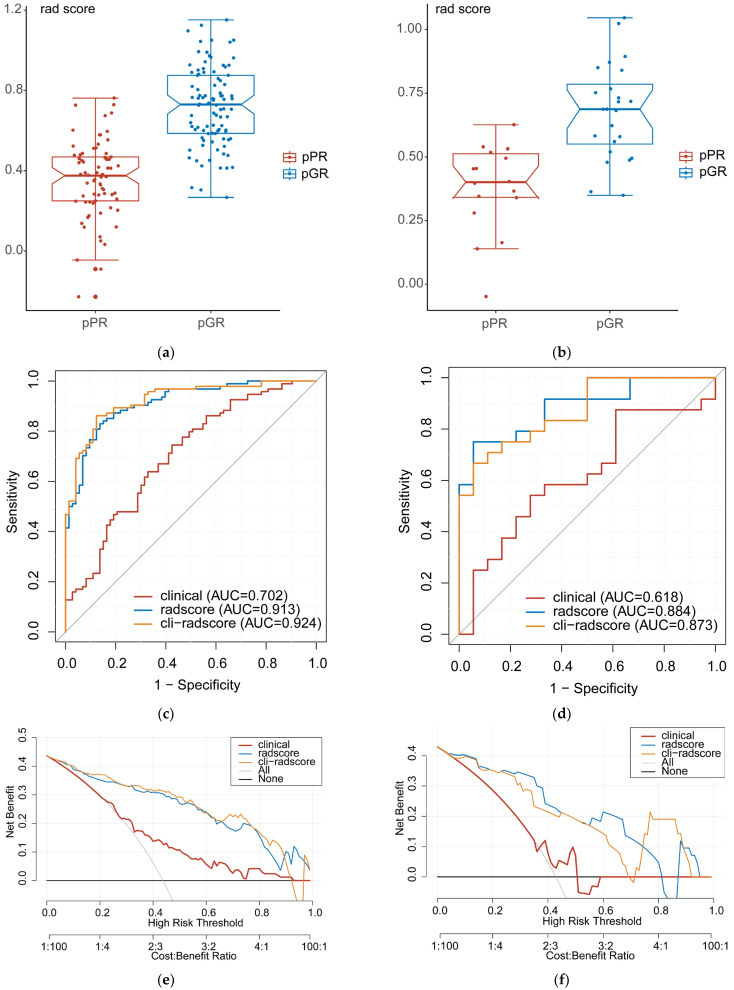
Scatterplots and boxplots of radscores for pGR and pPR groups in the training cohort (**a**) and validation cohort (**b**). The ROC curves of the clinical, radscore, and cli–radscore models for predicting pGR in the training cohort (**c**) and validation cohort (**d**). The decision curves for all models in the training cohort (**e**) and validation cohort (**f**). Notably, the decision curves for the validation cohort demonstrate that the predictive performance of the radscore model and cli–radscore model for pGR is comparable and superior to the clinical model at different threshold probabilities. The *y*–axis corresponds to the net benefit, with the gray line assuming all patients have pGR and the black line assuming all patients have pPR, while the *x*–axis represents the threshold probability. pGR, good pathological response; pPR, poor pathological response. ROC, receiver operating characteristic.

**Figure 6 diagnostics-13-01987-f006:**
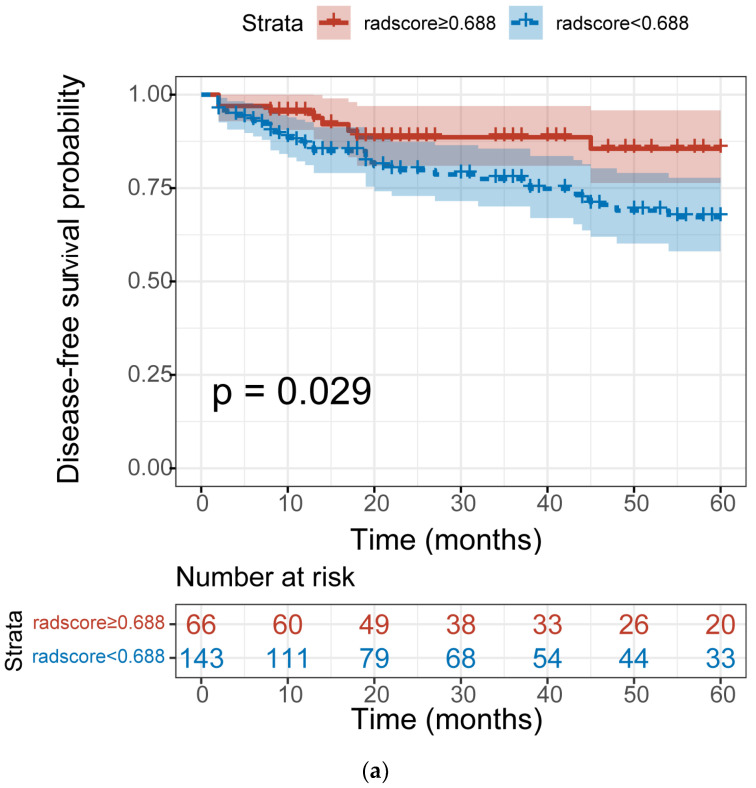
(**a**) Kaplan–Meier survival curves for disease–free survival by radscore. (**b**) The forest plot of the multivariable Cox regression results. HR, hazard ratio; MRF, mesorectal fascia involvement, EMVI, extramural vascular invasion; LPLN, lateral pelvic lymph node; CA199, carbohydrate antigen 199; pCR, complete pathological response.

**Table 1 diagnostics-13-01987-t001:** Patients’ characteristics.

Features	pPR (*n* = 91 ^a^)	pGR (*n* = 118 ^a^)	*p*
Gender			0.373
Male	67 (74%)	79 (67%)	
Female	24 (26%)	39 (33%)	
Age (years)	59.66 ± 11.83	60.32 ± 10.64	0.671
BMI (kg/m^2^)	24.39 ± 3.11	24.21 ± 3.17	0.682
cT			0.508
T_3_	79 (87%)	107 (91%)	
T_4_	12 (13%)	11 (9.3%)	
cN			0.771
N_0_	17 (19%)	22 (19%)	
N_1_	68 (75%)	85 (72%)	
N_2_	74 (7%)	96 (9%)	
DTAV			**<0.001**
≤4 cm	35 (38%)	19 (16%)	
>4 cm	56 (62%)	99 (84%)	
Tumor length			**0.049**
≤4 cm	30 (33%)	56 (47%)	
>4 cm	61 (67%)	62 (53%)	
MRF			**0.007**
Negative	50 (55%)	87 (74%)	
Positive	41 (45%)	31 (26%)	
EMVI			**0.029**
Negative	44 (48%)	76 (64%)	
Positive	47 (52%)	42 (36%)	
LPLN			0.698
Negative	64 (70%)	88 (74%)	
Positive	27 (30%)	31 (26%)	
CEA			0.647
≤5 ng/mL	51 (56%)	71 (60%)	
>5 ng/mL	40 (44%)	47 (40%)	
CA19–9			0.182
≤39 ng/mL	77 (85%)	108 (92%)	
>39 ng/mL	14 (15%)	10 (8.5%)	
WBC (×10^9^/L)	6.69 ± 1.80	6.30 ± 1.63	0.100
HGB (g/L)	136.73 ± 16.97	133.99 ± 21.19	0.315
PLT (×10^9^/L)	256.91 ± 70.17	233.87 ± 66.45	**0.016**
Lymphocyte (×10^9^/L)	1.74 ± 0.55	1.83 ± 0.59	0.278
Neutrophil (×10^9^/L)	4.34 ± 1.51	3.91 ± 1.38	**0.036**
Eosinophilic granulocyte (×10^9^/L)	0.16 ± 0.12	0.14 ± 0.10	0.100
Monocyte (×10^9^/L)	0.39 ± 0.15	0.41 ± 0.16	0.41
NLR	2.73 ± 1.44	2.37 ± 1.16	**0.047**
LMR	4.97 ± 2.17	4.97 ± 2.01	0.997
PLR	162.98 ± 77.89	139.78 ± 58.86	**0.015**
TRG			**<0.001**
0	0 (0%)	44 (37%)	
1	0 (0%)	74 (63%)	
2	71 (78%)	0 (0%)	
3	20 (22%)	0 (0%)	

^a^ Mean ± standard deviation; number (percentage). *p*–values < 0.05 are shown in bold. Abbreviations: pPR, poor pathological response; pGR, good pathological response; BMI, body mass index; cT, clinical T stage; cN, clinical N stage; MRF, mesorectal fascia involvement; EMVI, extramural vascular invasion; LPLN, lateral pelvic lymph node metastasis; CEA, carcinoembryonic antigen; CA199, carbohydrate antigen 199; WBC, white blood cell count; HGB, hemoglobin level; PLT, platelet count; NLR, neutrophil–to–lymphocyte ratio; LMR, lymphocyte–to–monocyte ratio; PLR, platelet–to–lymphocyte ratio; TRG, tumor regression grade.

**Table 2 diagnostics-13-01987-t002:** Performance of different models in the training and validation cohorts.

Models	Cohort	Cut–Off	AUC (95%CI)	F1 Score	Sensitivity	Specificity	Accuracy
LR_ITU_	Training	0.48	0.824 (0.764–0.887)	0.774	0.775	0.703	0.744
	Validation		0.779 (0.634–0.924)	0.741	0.728	0.705	0.718
LR_PTU_2mm_	Training	0.52	0.806 (0.74–0.873)	0.761	0.722	0.772	0.744
	Validation		0.785 (0.638–0.93)	0.758	0.72	0.769	0.742
LR_PTU_4mm_	Training	0.48	0.832 (0.776–0.899)	0.789	0.754	0.797	0.773
	Validation		0.79 (0.649–0.93)	0.749	0.712	0.757	0.732
LR_PTU_6mm_	Training	0.48	0.81 (0.745–0.877)	0.779	0.775	0.72	0.751
	Validation		0.79 (0.647–0.933)	0.772	0.762	0.725	0.746
LR_MR_F_	Training	0.50	0.723 (0.645–0.801)	0.745	0.822	0.5	0.682
	Validation		0.689 (0.522–0.854)	0.751	0.83	0.517	0.694
LR_MR_BVLN_	Training	0.54	0.789 (0.722–0.858)	0.695	0.614	0.802	0.696
	Validation		0.758 (0.61–0.906)	0.685	0.602	0.815	0.694
LR_ITU+PTU_2mm_	Training	0.49	0.831 (0.771–0.893)	0.779	0.754	0.764	0.758
	Validation		0.795 (0.654–0.935)	0.764	0.745	0.737	0.741
LR _ITU+PTU_4mm_	Training	0.50	0.832 (0.771–0.895)	0.778	0.714	0.843	0.77
	Validation		0.805 (0.667–0.944)	0.755	0.695	0.825	0.751
LR _ITU+PTU_6mm_	Training	0.51	0.874 (0.825–0.927)	0.807	0.801	0.761	0.783
	Validation		0.795 (0.659–0.931)	0.761	0.754	0.704	0.732
LR _ITU+MR_F_	Training	0.52	0.842 (0.784–0.903)	0.794	0.771	0.777	0.774
	Validation		0.79 (0.643–0.935)	0.745	0.712	0.748	0.727
LR _ITU+MR_BVLN_	Training	0.52	0.936 (0.904–0.972)	0.878	0.85	0.89	0.867
	Validation		0.859 (0.745–0.974)	0.811	0.789	0.803	0.794
LR _ITU+MR_F +MR_BVLN_	Training	0.55	0.873 (0.825–0.926)	0.782	0.705	0.874	0.779
	Validation		0.85 (0.736–0.967)	0.763	0.679	0.879	0.766

**Table 3 diagnostics-13-01987-t003:** Univariate and multivariate Cox analysis of 5–year disease–free survival in locally advanced rectal cancer after surgery.

Factor	Number (*n*)	Univariate Cox Analysis			Multivariate Cox Analysis	
		HR (95%CI)	*p*	C–Index	HR (95%CI)	*p*
Tumor length			**0.016**	0.585		0.53
≤4 cm	86	1				
>4 cm	123	2.328 (1.144–4.736)			1.304 (0.57–2.985)	
MRF			**0.018**	0.591		0.356
Negative	137	1				
Positive	72	2.05 (1.119–3.755)			0.704 (0.335–1.482)	
EMVI			**<0.001**	0.678		**0.009**
Negative	120	1				
Positive	89	4.102 (2.098–8.018)			2.799 (1.292–6.062)	
LPLN			**<0.001**	0.636		**0.015**
Negative	151	1				
Positive	58	3.42 (1.863–6.281)			2.251 (1.17–4.332)	
cT			**0.036**	0.555		0.654
T3	186	1				
T4	23	2.245 (1.036–4.862)			1.211 (0.524–2.801)	
CA19–9			**0.014**			0.234
Negative	185	1				
Positive	24	2.461 (1.177–5.146)			1.629 (0.73–3.638)	
radscore			**0.034**	0.576		0.78
≥0.688	52	1				
<0.688	157	2.303 (1.065–4.977)			1.126 (0.489–2.595)	
pCR			**0.007**	0.59		0.103
No	165	1				
Yes	44	0.176 (0.043–0.728)			0.294 (0.068–1.279)	

*p*–values < 0.05 are shown in bold. Abbreviations: HR, hazard ratio; MRF, mesorectal fascia involvement, EMVI, extramural vascular invasion; LPLN, lateral pelvic lymph node; CA199, carbohydrate antigen 199; pCR, complete pathological response.

## Data Availability

Not applicable.

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
