# Peer review of "Radiomics from Mesorectal Blood Vessels and Lymph Nodes: A Novel Prognostic Predictor for Rectal Cancer with Neoadjuvant Therapy"

_diagnostics, 2023, doi:10.3390/diagnostics13121987_

Round 1

Reviewer 1 Report

This manuscript discuss about the “Radiomics from Mesorectal Blood Vessels and Lymph Nodes:  A Novel Prognostic Predictor for Rectal Cancer with Neoadjuvant Therapy”. An interesting knowledge has been reported. However the following comments should be addressed before acceptance

Major comments

1.      Novelty of the manuscript must be better emphasized

2.      In abstract section: there no significant statement regarding the importance of this study

3.      A clear conclusion statement (one line) should be given at the end of the abstract section

4.      Keywords are not catchy

5.      The importance of this study should be clealry given in the introduction part- add how Radiomics useful for cancer especially colorectal cancer

6.      Author should discuss about how significant differences of platelet and neutrophl count influence the features

7.      What is the novelty of Figure 5.a and b

8.      How does The follow-up period effects the analysis

9.      I appreciate the authors limitation statement about their study

10.  Suggested to add some interesting results/findings in conclusion section

11.  Authors should discuss more about the characteristic feature, uniquesness and ddifference of clinical, Radscore and Cli-radscore models

12.  Enhance the quality of images, especially Figure 1, 5 e and f, 6a and b

After addressing all the comments, this manuscript can be acceptable for further comment

Author Response

Dear Reviewer,

Thank you for taking the time to evaluate our manuscript entitled “Radiomics from Mesorectal Blood Vessels and Lymph Nodes: A Novel Prognostic Predictor for Rectal Cancer with Neoadjuvant Therapy” and providing insightful comments. We have carefully addressed each of your points and believe that our revisions have significantly improved the manuscript. Please find below our responses to your comments:

  1. Novelty: We have emphasized the novelty of this article in the third paragraph of the introduction, emphasizing how our research contributes to the existing knowledge.

  1. Abstract: We have underscored the significance of this study in the final sentence of the abstract.

  1. Abstract conclusion: A clear, concise conclusion statement has been added at the end of the abstract.

  1. Keywords: We have modified the keywords to be more relevant and catchy.

  1. Importance of the study in the introduction: We have incorporated the application of radiomics in colorectal cancer in the second and third paragraphs of the abstract.

  1. Significance of platelet and neutrophil count: We have incorporated the role of clinical factors in the model in the first paragraph of the discussion and detailed the results of univariate and multivariate analyses on the training set in Table A17. As the platelet count and neutrophil count were not significant in the multivariate analysis, we did not discuss these two factors specifically, but rather elaborated on their instability.

  1. Novelty of Figure 5.a and b: We have provided a detailed explanation of the novelty of these figures in the figure legend.

  1. Follow-up period: We have elucidated the follow-up duration and the number of positive events for each group, classified based on radscore, in line 375.

  1. Limitation statement: Thank you for your positive feedback.

  1. Conclusion: We have revised the conclusion section to include more interesting findings from our study.

  1. Discussion of models: We have expanded our discussion about the unique characteristics and differences of the clinical, Radscore and Cli-radscore models in the first paragraph of the discussion.

  1. Image quality: We have worked to enhance the quality of the images, especially Figure 1, 5 e and f, 6a and b.

We sincerely appreciate your constructive comments which have helped us greatly improve our manuscript. We believe that after these revisions, our manuscript will be acceptable for publication.

Thank you once again for your valuable comments.

Best Regards,

Ning Lang.

Reviewer 2 Report

Dear Editor,

I have reviewed the manuscript " Radiomics from Mesorectal Blood Vessels and Lymph Nodes: A Novel Prognostic Predictor for Rectal Cancer with Neoadjuvant Therapy". Qin et al. aimed to develop a radiomic model combining intratumoral and peritumoral features to predict good pathological response (pGR) to neoadjuvant chemoradiotherapy (nCRT) in patients with locally advanced rectal cancer (LARC). A very good study was conducted with 209 cases. I congratulate the authors for this work that will attract the attention of the reader.

Best regards.

Minor editing of English language required

Author Response

Dear Reviewer,

We sincerely appreciate your time and effort in reviewing our manuscript "Radiomics from Mesorectal Blood Vessels and Lymph Nodes: A Novel Prognostic Predictor for Rectal Cancer with Neoadjuvant Therapy". We are encouraged by your positive feedback and recognition of the value and interest of our study.

We believe our work contributes to the current understanding of utilizing radiomics in predicting good pathological response to neoadjuvant chemoradiotherapy in patients with locally advanced rectal cancer. The encouragement such as yours motivates us to continue our research in this important area.

Thank you once again for your kind words and support.

Best regards,

Ning Lang

Reviewer 3 Report

In the article "Radiomics from Mesorectal Blood Vessels and Lymph Nodes: A Novel Prognostic Predictor for Rectal Cancer with Neoadjuvant Therapy", Qin et al. propose an innovative method to pre-therapeutically predict the pathological response after neoadjuvant treatment by chemoradiotherapy in locally advanced rectal cancer. The authors approach the radiomic analysis differently, going beyond the classic concepts involving ROIs from tumor and peritumoral regions. The study proposes and identifies the integration of radiomic features from intratumoral, and mesorectal blood vessels+ lymph nodes ROIs along with clinical and biological features. The article is rigorously documented and written very explicitly, it also presents in detail the method of analysis and construction of radiomics models and scores, presenting in 3 tables the characteristics of the patient, the analysis of the performance of the proposed models and univariate and multivariate Cox analysis of 5-year disease-free survival in locally advanced rectal cancer after surgery. The conclusions are interesting and multiple: the identification of distance from the mass to the anal verge (DTAV) and platelet-to-lymphocyte ratio as independent predictors of the pathological response, but especially the lack of benefit incorporating clinical factors in predictive ability and clinical applicability of the radscore. The presentation of the graphs is detailed and sometimes difficult to interpret for a clinician. I mention here ICC boxplots and violin plots for each ROI, suggestive methods, but which require a high level of understanding of statistical analysis. The project can be considered a reference in emphasizing the importance of the mesorectal vessels and lymph nodes, not only in the particular case of the mesorectum, but also in general in the radiomics evaluation of these structures from tumor proximity. Without having deep expertise in mathematical and statistical methods, as a radiation oncologist I also recommend the evaluation of thearticle by an expert in mathematical models. Another strong point is the very pertinent identification of the limits of the study. Overall, I recommend publishing the article

Author Response

Dear Reviewer,

We appreciate your thorough review of our manuscript, "Radiomics from Mesorectal Blood Vessels and Lymph Nodes: A Novel Prognostic Predictor for Rectal Cancer with Neoadjuvant Therapy". Your detailed feedback is invaluable to us and we are encouraged by your recognition of the novelty and rigorousness of our study.

As suggested, we will seek input from an expert in mathematical models to ensure the robustness of our analysis. We are grateful for your recommendation to address this aspect.

We appreciate your positive feedback on the identification of the limitations of our study. We believe it's crucial to acknowledge these aspects to ensure our findings are interpreted accurately and contextually.

Once again, thank you for your supportive comments and recommendation to publish our manuscript. We look forward to further improving our work based on your valuable insights.

Best regards,

Ning Lang

Reviewer 4 Report

This paper reports that the radiomic features obtained from tumoral and peritumoral regions of interest are suitable for predicting good pathological response to neoadjuvant chemoradiotherapy in people suffering from locally advanced rectal cancer.   The study is statistically sound.  The introduction does an adequate effort to review the field.  The methods described could be reproducible but are dependent on the patients tested.   The results are clearly presented.  The discussion is within the framework of the current literature, and the research presented.  The conclusion is supported by the results. 

All the cited references are relevant to the research presented.  However some concepts can benefit from expansion, such as the statically methods employed in the report.

The research is appropriately designed and implemented.  This reviewer was left wondering of the limitations of the sample size of the study.

The results are clearly presented, but the figures need improvement (see below). 

Specific comments to be addressed prior to publication:

1)       The authors could cite and review similar studies which use the method of validation applied in this study.  This will allow readers to better understand the merits and limitations of the validation methods in this paper.

2)      The Figures are excellent and clearly explain the concepts.  However the text in the plots on the axis is impossible to read.  The plots should be redone so the text is at a readable size for the audience of the journal.

3)      The text in the diagram in Figure 1 is very difficult to read.

OK.

Author Response

Dear Reviewer,

We greatly appreciate the time and effort you put into reviewing our manuscript, "Radiomics from Mesorectal Blood Vessels and Lymph Nodes: A Novel Prognostic Predictor for Rectal Cancer with Neoadjuvant Therapy". Your thoughtful comments and constructive feedback have helped us further enhance the quality of our work.

1) As you suggested, we have now added a thorough review and citation of studies that used similar validation methods in the second paragraph of discussion. This should allow the readers to better grasp the strengths and limitations of our validation approach.

2) The figures have been revised to ensure better readability. The text in the plots on the axis has been resized to be easily readable by the journal's audience.

3) We have also made the text in Figure 1 diagram larger and clearer to read.

We are confident that these revisions have improved the manuscript and addressed your concerns effectively. We sincerely appreciate your insightful comments and look forward to your continued feedback.

Sincerely,

Ning Lang.

Round 2

Reviewer 1 Report

Accept

Author have addressed all the comments and thus enhances the quality of the manuscript